# 5-Fluorouracil Conversion Pathway Mutations in Gastric Cancer

**DOI:** 10.3390/biology9090265

**Published:** 2020-09-02

**Authors:** Alessio Biagioni, Fabio Staderini, Sara Peri, Giampaolo Versienti, Nicola Schiavone, Fabio Cianchi, Laura Papucci, Lucia Magnelli

**Affiliations:** 1Department of Experimental and Clinical Biomedical Sciences “Mario Serio”, University of Florence, Viale G.B. Morgagni 50, 50134 Firenze, Italy; giampaolo.versienti@unifi.it (G.V.); nicola.schiavone@unifi.it (N.S.); laura.papucci@unifi.it (L.P.); lucia.magnelli@unifi.it (L.M.); 2Department of Experimental and Clinical Medicine, University of Florence, Largo Brambilla 3, 50134 Firenze, Italy; fabio.staderini@unifi.it (F.S.); sara.peri@unifi.it (S.P.); fabio.cianchi@unifi.it (F.C.)

**Keywords:** 5 fluorouracil, chemoresistance, gastric cancer, mutations

## Abstract

To date, 5-Fluorouracil (5FU) is a major component of several chemotherapy regimens, thus its study is of fundamental importance to better understand all the causes that may lead to chemoresistance and treatment failure. Given the evident differences between prognosis in Asian and Caucasian populations, triggered by clear genetic discordances and given the extreme genetic heterogeneity of gastric cancer (GC), the evaluation of the most frequent mutations in every single member of the 5FU conversion and activation pathway might reveal several important results. Here, we exploited the cBioPortal analysis software to query a large databank of clinical and wide-genome studies to evaluate the components of the three major 5FU transformation pathways. We demonstrated that mutations in such ways were associated with a poor prognosis and reduced overall survival, often caused by a deletion in the *TYMP* gene and amplification in *TYMS*. The use of prodrugs and dihydropyrimidine dehydrogenase (DPD) inhibitors, which normally catabolizes 5FU into inactive metabolites, improved such chemotherapies, but several steps forward still need to be taken to select better therapies to target the chemoresistant pools of cells with high anaplastic features and genomic instability.

## 1. Introduction

The drug 5-fluorouracil (5FU) is one of the most commonly used drugs to treat several kinds of tumors, including breast cancer, head and neck cancers, gastric cancer, colon cancer, and melanoma [1]. Discovered for the first time in 1957 by Charles Heidelberger and colleagues at the University of Wisconsin [2], it has become a major therapeutic option when specific targets are not available and when surgery is not practicable. Indeed, 5FU is actively absorbed by highly proliferating tissues, especially tumor ones, that need uracil for nucleic acid biosynthesis, and thus acts by inhibiting tumor cell division exploiting two different ways: it may inhibit the thymidylate synthase (TYMS), forming a covalent ternary complex with 5,10-methylenetetrahydrofolate, suppressing in that way the DNA synthesis, or it might be incorporated into RNA molecules blocking gene expression [3]. Once internalized into tumor cells, 5FU is converted to fluorouridine monophosphate (FUMP) by the orotate phosphoribosyltransferase (OPRT) and then converted to fluorodeoxyuridine monophosphate (FdUMP) by the ribosyl reductase complexes (RRM1/2), a biochemical process better known as the “OPRT–RR pathway” [4]. While FdUMP inhibits DNA synthesis through its inhibitory action on TYMS, FUMP is largely responsible for the inhibition of gene expression through its incorporation into mRNAs. Another conversion way is mediated by the action of the thymidine phosphorylase (TYMP), which is able to convert 5FU into fluorodeoxyuridine (FdU) and then transformed to FdUMP due to the action of the thymidine kinase 1 (TK1), in a process called “TP–TK pathway”. FdUMP might also be converted back to FdU by the nucleotidase NT5E [5]. Moreover, uridine monophosphate synthase (UMPS) and phosphoribosyl pyrophosphate amidotransferase (PPAT) are also responsible for the conversion of 5FU to FUMP, which is in turn processed to fluorouridine diphosphate (FUDP) and then to FdUMP by RRM1 and 2 [6,7]. FUMP reservoir can be fueled also by the transformation of fluorouridine (FUR) triggered by uridine kinase (UK), generated in turn by the conversion of 5FU by the uridine phosphorylase (UP). All these steps are summarized in the schematic view in Figure 1.

In order to study chemoresistance and its side-effects, it is important to remark the action of the dihydropyrimidine dehydrogenase (DPD), which catabolizes 5FU into inactive metabolites by the liver [8], producing fluoroacetate and fluorohydroxypropionic acid, which have been reported to induce cardiotoxicity and neurotoxicity [9,10]. While 5FU is normally administered intravenously to avoid the DPD digestion by the gut wall, the use of oral 5FU prodrugs (capecitabine, tegafur, and doxifluridine) [11,12,13] and DPD inhibitors (gimeracil, uracil, eniluracil, 5-chloro-2,4-dihydroxypyridine) [14,15,16] partially resolves the above-described side-effects. Every single mutation in one of the above-described enzymes might affect irreversibly the sensitivity to 5FU leading to the generation of subpopulations of chemoresistant cancer cells, which cause therapy failure. Indeed, gastric cancer (GC), together with bladder, lung and melanoma, is one tumor histotype with the highest mutation frequency [17]. GC is to date the fifth most frequently diagnosed cancer and the third leading cause of cancer death according to GLOBOCAN 2018 [18], and due to the lack of macroscopical manifestations in the early phases of the disease, it is often diagnosed only after metastasis in advanced stages, when surgery is not practicable and the only therapeutic option is chemotherapy. According to the international guidelines, the first-line approach for therapy would involve treating HER2 negative patients with platinum-based drugs (cisplatin or oxaliplatin) and fluoropyrimidines (5-fluorouracil or capecitabine) otherwise, trastuzumab is added to treat HER2 positive patients. In Eastern countries, S-1 (tegafur-gimestat-otastat potassium), an oral fluoropyrimidine regimen, is commonly used [19]. The addition of taxanes to first-line chemotherapy, as in DCF (docetaxel, cisplatin, and 5-fluorouracil) or FLOT (5-fluorouracil, oxaliplatin, and docetaxel) is normally exploited only for young patients with good performance due to high toxicity effects [20,21]. To date, the second-line regimens are composed of ramucirumab alone or in combination with paclitaxel [22,23] or 5FU and irinotecan (FOLFIRI) [24] when patients are not responsive to the first ones. The vast majority of GC are adenocarcinomas, which can be classified mainly by two different approaches: the Lauren histopathology system, which distinguishes the intestinal, the diffuse and the mixed subtypes based on immunohistochemical analysis, and the Cancer Genome Atlas (TCGA) research network-derived classification, based on in situ hybridization techniques dividing GC into four subgroups, i.e., Epstein–Barr virus-positive, microsatellite unstable, genomically stable, and chromosomally unstable GCs [25,26]. In this context, another molecular classification called “Singapore–Duke” was recently reported [27]. Such a study analyzed the gene expression profiles of 248 gastric tumors, identifying three different subtypes (proliferative, metabolic, and mesenchymal) [21]. The proliferative subtype was characterized by high expression of *E2F*, *MYC*, *RAS*, cell cycle genes and frequent *TP53* mutation, copy number amplification, DNA hypomethylation, and a Lauren intestinal type, while the metabolic was reported to gain an upregulation of metabolic and digestion-related genes that are normally expressed in gastric mucosa, elevated expression of cell adhesion proteins, extracellular matrix (ECM) receptors, and the activation of EMT and cancer stem cell pathways. The mesenchymal subtype was instead associated with a Lauren diffuse type, characterized by alterations in *TP53*, transforming growth factor β (TGFβ), vascular endothelial growth factor (VEGF), NF-κB, mTOR, and Shh signaling pathways. It is relevant to report that the Singapore–Duke study evidenced that the metabolic subtype was more sensitive to 5FU than the others, including metabolic GC patients that showed greater responsiveness to 5FU chemotherapy. Such higher sensitivity could depend on the reported lower expression of TYMS and DPD proteins in the metabolic GCs; however, the three subtypes showed no significant differences in cancer-specific and disease-free survival. Indeed, the new molecular classifications were introduced due to the extreme genetic heterogeneity of GC that has also been observed in studies on somatic copy number alterations, gene mutations, and epigenetic and transcriptional changes [28]. Moreover, in accordance with the elevated genetic instability, mutations found in GCs are extremely divergent and, actually, no characteristic driver gene mutations have been identified. Here, we focused on the most recurrent gene mutations in cancer, i.e., amplification, missense, and truncating mutations, frameshift effects, and oncogenes fusion. We analyzed the major steps in 5FU conversion, evaluating and discussing where worldwide research need to focus the attention for future therapies. Our manuscript aims to focus the attention of the scientific community on the research of reliable markers, which might reflect the chemosensitivity status of the patients and may be evaluable in course of therapy.

## 2. Materials and Methods

### 2.1. Analysis of Genomic Data

The cBioPortal for Cancer Genomics [29] was used to explore, visualize, and analyze cancer genomics data from human cancer tissues and cell lines. The database was queried for UPP1, UCK1, UMPS, RRM1/2, TYMP, TYMS, TK1, NT5E, and PPAT, and the conditions for the analysis visualization were adjusted as reported in each legend. The genomic data types evaluated include genomic mutations, somatic mutations, and DNA copy-number alterations (CNAs). The effect of target genes mutation on the prognosis of GC was evaluated by a Kaplan–Meier plotter, generated by cBioPortal tool (https://www.cbioportal.org/).

### 2.2. Protein–Protein Interaction Analysis

STRING v11 was used to generate and visualize a complex map of all the known and predicted interactions among the queried proteins setting 0.400 as the minimum required confidence for the interaction score [30]. The proteins involved in angiogenesis were selected from the most recent topic publications available in the literature. 

### 2.3. TYMP Expression Analysis

The expression level of TYMP in GC vs. normal tissues was analyzed through the “cancer vs. normal” filter in the Oncomine database selecting all GC studies available [31,32]. All data conforming to the criteria of *p* < 0.05, fold-change > 2, and a gene rank percentile <10% were included in the present study [33].

### 2.4. Statistical Analysis

Survival curves and graphs were plotted using cBioPortal, displayed with p values calculated by long-rank test. The statistical analysis of the data retrieved from Oncomine was calculated by *t*-test. All values were corrected on the false discovery rate.

## 3. Results

### 3.1. 5FU System Mutations Lead to a Poor Prognosis

As 5FU is one of the most diffuse components of chemotherapy regimens in many kinds of cancers, we decided to exploit the cBioPortal analysis software and database to better understand the impact of mutations in any of the enzymes that belong to such a pathway. We identified 10 genes, as reported in the Material and Methods Section, involved in 5FU conversion, and we selected two cohorts: one comprising all patients with at least one of the selected genes mutated (the altered group) and one including all with “unaltered” genes. Evaluating all kind of mutations in a broad range of cancer studies (32 studies including 10953 patients, TCGA PanCancer Atlas), we observed significant changes in the overall survival (OS) curve, with a reduction of the median OS time from 80.74 months of the unaltered group to 67.46 months of the altered one (Table 1), as shown in the Kaplan–Meier plots in Figure 2. We also evidenced a lower median for disease-free (DF), progression-free (PFS), and disease-specific survival (DSS) in the altered cohort. Such indexes were calculated as the disease-free status since the initial treatment, the chance of staying free of disease progression after treatment, and the percentage of patients who have not died starting at the time of diagnosis or at the start of treatment, respectively. In Table 1, we indicated the percentage and the median of such indexes. It is important to remark that we were unable to calculate the median month DF as the curves did not interpolate the 50% of the y-axis, but nonetheless, the difference between the two groups was statistically significant (*p* = 0.0114). Using an alternative analysis strategy, we also compared the unaltered cohort to the singular mutation genes groups evidencing that while mutations in RRM1 might have a positive impact on the OS, mutations in UPP1 are strongly associated with a worse prognosis (Appendix A). Larger studies comprising not only the mutational profile but also a proteomic analysis will better clarify the real impact of such mutations in cancer progression.

### 3.2. 5FU Conversion System Mutation Frequency among Cancers

To better understand the relevance of 5FU conversion system mutations in cancers, we analyzed the frequency of the 10 queried genes in several types of cancers, taking into consideration all kinds of possible mutations. Globally, such genes were found to be altered in a total of 1488 (14%) of queried samples/patients, including a total of 639 different genetic alterations. As shown in Figure 3, the three most frequently mutated histotypes of cancers are the ovarian, the endometrial, and the bladder urothelial carcinoma, while esophagogastric cancers are only ninth in such classifications. While the proportions among the several kinds of mutations are almost constant, it is evident that for ovarian cancer and prostate adenocarcinoma deep deletions account for about half of the analyzed mutations.

### 3.3. Individual Analysis of Mutation Frequency

More than a global analysis, the queried genes were analyzed also for mutation frequency singularly. As shown in Figure 4, TYMP had the highest percentage of mutation, especially in ovarian epithelial cancer, while UCK1, which is frequently mutated in endometrial carcinoma, is the less present among the screened genes. Indeed, UCK1 is not often observed mutated and, as already evaluated by Murata and colleagues, its level did not change even in a chemoresistant model of human fibrosarcoma and gastric carcinoma [34]. It is interesting to notice that the vast majority of mutations for TYMP are classified as deep deletions, while for TYMS they are amplifications. These phenomena could be in line with the fact that TYMP is responsible for the conversion of 5FU into its active form, while TYMS is its final target, responsible for the conversion of deoxyuridine monophosphate (dUMP) to deoxythymidine monophosphate (dTMP), an important step to fuel the pool of DNA precursor for replication. Thus, when TYMP is deleted 5FU could only be activated by the other two pathways (see Introduction) leading to worse chemotherapy response and to the increase of the concentration of inactive 5FU [35] causing cytotoxic side-effects, while amplification of TYMS makes harder for 5FU to exert its action [36].

### 3.4. 5FU Conversion System Mutations Analysis in GC

As reported by Lawrence et al. [17], gastrointestinal tumors demonstrated an extremely high frequency of transition mutations at CpG dinucleotides, and also Tan et al. [28] stated that GCs are often characterized by several somatically acquired mutations in various genes, estimating approximately 50 to 70 nonsynonymous mutations, a mutation level comparable to colon and esophageal cancers. Indeed, while previously GC was believed to rely on mutations induced only by environmental factors (food and alcohol habits, smoke, BMI, and Helicobacter pylori), recently it was better understood the predominant role of the genetic compartment, as demonstrated by the genetic and molecular discordances between different race/ethnicities [37] or sex (i.e., the signet cell ring carcinoma—[38]). Moreover, to date, being well-known the discordances between Asian and Caucasian populations’ prognosis in GC [37,39], it is now clear that the genetic component exerts a fundamental action especially in this kind of cancer. To better understand the role of the 5FU conversion system mutations in GC, we analyzed all the above-described genes focusing on the mutations presented only in GC studies [26,29,40,41,42,43,44,45,46,47,48]. As shown in Table 2, the most frequently mutated genes were TYMP, NT5E, UPP1, and UCK1, whereas globally the missense mutations were predominant with respect to the truncating ones, while inframe mutations were reported only in NT5E. We also reported in the Appendix A, the number of cases of co-occurrence of 5FU-related mutations, highlighting that most patients in our study suffered from only one mutation at a time among the queried ones.

### 3.5. The Paradoxical TYMP Expression in GC

It is well-known that TYMP plays a fundamental role in the transformation of 5FU to its active compound form, and we reported to have the highest mutation frequency (2.8%), accounting for four different missense mutations, as shown in Figure 5.

As reported above, TYMP is often deleted in many kinds of cancers but not in GC. Indeed, even though the loss of TYMP expression leads to an unsuccessful 5FU conversion, causing inevitably the treatment failure and severe cytotoxic effects due to its accumulation, its expression was found to be significantly higher in GC tumor tissues, creating a paradox. We decided to examine two studies reporting TYMP expression comparing GC to the normal gastric epithelia through Oncomine software [31,32]. As shown in Figure 6, TYMP expression is low in the gastric epithelial mucosa, while it is upregulated in all the different GCs.

Such a phenomenon was also reported by Kimura et al. who described TYMP as a powerful prognostic tool [49]. Indeed, evaluating TYMP expression through the ELISA (enzyme-linked immunosorbent assay) technique, they examined 263 samples from patients who underwent gastrectomy and identified a correlation between TYMP and the metastatic process, reporting a high TYMP expression in the tumor-invading serosa and tumor tissues from patients with lymph node metastasis and lymphatic invasion. Moreover, Tabata et al. reported that TYMP expression is higher in tumor tissues than in adjacent non-neoplastic ones in several kinds of cancers. They also correlated its expression with poor prognosis in colon and differentiated gastric carcinomas, and they demonstrated that thymidine can also serve as a substrate for the glycolytic pathway in human cancer cells [50]. We do believe that future data about 5FU chemoresistant patients might improve our knowledge about TYMP as a prognostic marker.

### 3.6. 5FU Conversion System Is Associated with Tumor Angiogenesis

Each member of the 5FU conversion system plays a specific role in the activation of the chemotherapeutic agent, but such a mechanism is not exclusively associated with the pyrimidine metabolism. Indeed, the proteins codified by the genes examined in this article are closely associated with angiogenesis. As shown in the map in Figure 7A, we identified several interactions among 5FU converting pathway enzymes and some proteins that are frequently associated with the angiogenesis biological process. Most of the connections were linked through NT5E, which has already been described as a master angiogenesis regulator [51], and also TYMP plays a fundamental role, interacting with PECAM1, KDR, FGF2, VEGFA, and ANGPT2. Even though they interact actively, their expression is not always correlated in GC. Exploiting cBioPortal analysis software, we reported the mutation frequency of the above-described genes in the altered and unaltered cohorts, comprising 211 and 1154 patients, respectively. As shown in Figure 7B, only ITGB1, ITGA5, PDGFRB, KDR, ANGPT1/2, and Notch1 were found to be mutated more frequently in association with the alterations of the 5FU converting system. We also revealed that mutations in the 5FU conversion system are correlated with a higher mutation frequency of KLF5, Ki67, and CAIX, which are typical progression-associated markers in GC, as shown in Appendix A.

## 4. Discussion

5FU is a major component of several chemotherapy regimens, exploited to treat a multitude of different cancers. Even though 5FU use had improved in a significant way, the patients’ OS, its cytotoxic effects as well as the high probability to gain chemoresistance make it necessary to study further its metabolic and catabolic pathway. We analyzed, exploiting several bioinformatic tools, the impact and the frequency of mutations in each pathway of the 5FU metabolic biological processes. We demonstrated that such mutations were associated with a poor prognosis, with a reduced OS, DF, PFS, and DSS. For this reason, we do believe that such alterations should be constantly evaluated in patients, especially exploiting the so-called “liquid biopsy” [52]. Indeed, once patients underwent surgery, oncologists are guided by the Lauren and the TCGA classifications, but during the therapy, there are not available options to gain useful information about the disease status and to monitor the success of the treatment. Therefore, the combination of a new panel of cancer progression-associated markers and the liquid biopsy might improve the knowledge of the tumor characteristics and composition, gaining crucial data about the chemosensitivity status of patients, supporting in such a way the best choice of treatment. We also identified the ovarian, endometrial, and bladder urothelial carcinomas as the cancers histotype with the highest mutation frequency, while esophagogastric cancers account for more than 15% of mutation frequency, almost equally divided by missense, deletion, and amplification. Among them, we reported *TYMP* to be often deleted in several kinds of cancers but not in GC, while *TYMS* is frequently amplified, which may lead to treatment failure [35,36]. However, more data are needed to better understand their roles: as reported, the deep deletion is the most common *TYMP* mutation among cancers, leading to a complete loss of expression, while in GC we reported four missense mutations. Therefore, such deletion of *TYMP* results in a rare event in GC and needs to be verified on more samples, as its expression is reported to be increased in both diffuse and intestinal gastric carcinoma compared to the epithelial mucosa. Moreover, *TYMP* is commonly reported to be associated with the metastatic phenotype [49], but no statistical relevant data are available about its expression in chemoresistant patients. To date, data from literature are only based on in vitro experimentations on cell lines chronically exposed to 5FU, but the evaluation of tissue samples from patients is needed to further examine this phenomenon [53]. In contrast with the reported expression data, Kawahara et al. described *TYMP* only in tumor-associated macrophages and not in GC cells but limiting the study only to Asian- and African-derived GC cell lines [54]. Such *TYMP*-positive macrophages were demonstrated to promote angiogenesis and metastasis in GC. Its association with the most commonly reported angiogenic proteins is another point that will be faced with additional experimental tools. Indeed, we verified that in particular *TYMP* and the Ecto-5′-nucleotidase *NT5E* constitute crossing points with angiogenesis and thus their unbalanced expression in GC may lead to an improved tumor vascular formation or to the stimulation of the vasculogenic mimicry process [55,56,57]. Such unbalanced expression in GC, altering the angiogenic vessel network and triggering a vascular mimicry response, might be one possible phenotype of the chemoresistant pool of cells that often leads to treatment failure [58]. It is interesting to mention the mitochondrial neurogastrointestinal encephalomyopathy (MNGIE), which is a rare multisystemic autosomal recessive disorder caused by *TYMP* mutations [59]. Such mutations generate a severe thymidine phosphorylase deficiency leading to ptosis, progressive ophthalmoplegia, gastrointestinal dysmotility, cachexia, peripheral neuropathy, and leukoencephalopathy. The MNGIE has a higher frequency in European countries with typical mutation profiles, which might be family inherited and may hide a possible relationship with cancer. We need to point out that our analysis is limited due to the lack of information about the race/ethnicity, which may influence gene expression and mutations pattern [37]. Finally, we can affirm that the understanding of the molecular mechanisms regulating 5FU activation and the consequences of its chronic use will need to be further evaluated by analyzing biopsy samples and by generating chemoresistant cells, taking into consideration all the possible variables, such as race/ethnicity and sex.

## 5. Conclusions

Due to the lack of useful prognostic powerful tools, the selection of efficient chemotherapies is a hard journey for clinicians. When a patient underwent gastrectomy, the Lauren and the TGCA classifications describe only how to stratify such tumor masses but rarely give information on which way needs to be walked to achieve a successful treatment. HER2 evaluation might shed light on the use of Trastuzumab, but few other options are currently available as “targeted” therapies. We do believe that the evaluation of *TYMP*, as the key enzyme of the 5FU pathway, and being the one which is often find mutated in GC might improve the choice for the best therapy regimen. Indeed, the use of a wrong treatment not only causes cytotoxicity without any beneficial effects but might also select a pool of chemoresistant cells that, stimulating tumor angiogenesis, may gain a selective advantage to access blood and lymphatic vessels to metastasize by stimulating tumor angiogenesis.

## Figures and Tables

**Figure 1 biology-09-00265-f001:**
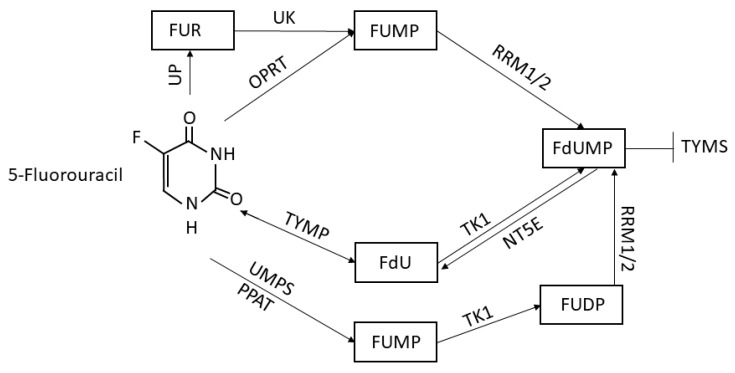
A simplified diagram of the 5-fluorouracil (5FU) conversion system. Uridine phosphorylase (UP), uridine kinase (UK), fluorouridine (FUR), orotate phosphoribosyltransferase (OPRT), fluorouridine monophosphate (FUMP), ribosyl reductase complexes (RRM1/2), fluorodeoxyuridine monophosphate (FdUMP), thymidylate synthase (TYMS), thymidine phosphorylase (TYMP), fluorodeoxyuridine (FdU), thymidine kinase 1 (TK1), nucleotidase (NT5E), uridine monophosphate synthase (UMPS), phosphoribosyl pyrophosphate amidotransferase (PPAT), and fluorouridine diphosphate (FUDP).

**Figure 2 biology-09-00265-f002:**
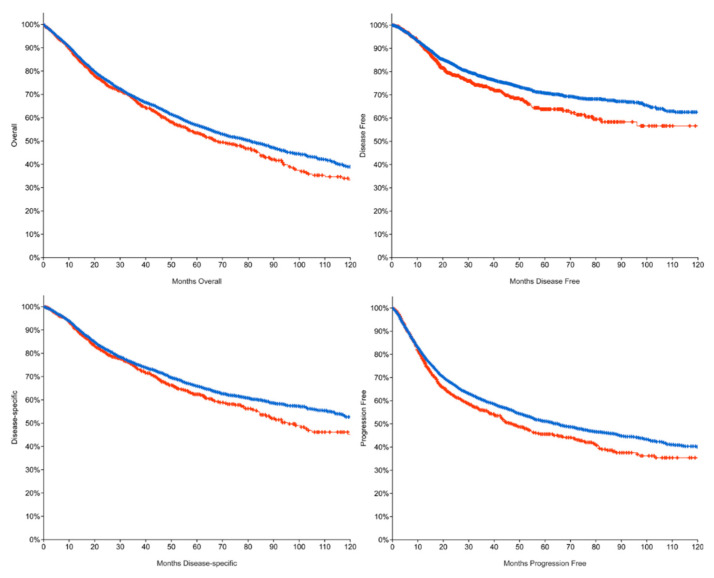
Red: altered group; blue: unaltered group; we observed that mutations in the 5FU converting system influenced the overall survival (*p* = 0.0163), the disease-free survival (*p* = 0.0114), the progression-free (*p* = 0.000197), and the disease-specific curves (*p* = 0.00679); data retrieved from 32 studies including 10953 patients, TCGA PanCancer Atlas.

**Figure 3 biology-09-00265-f003:**
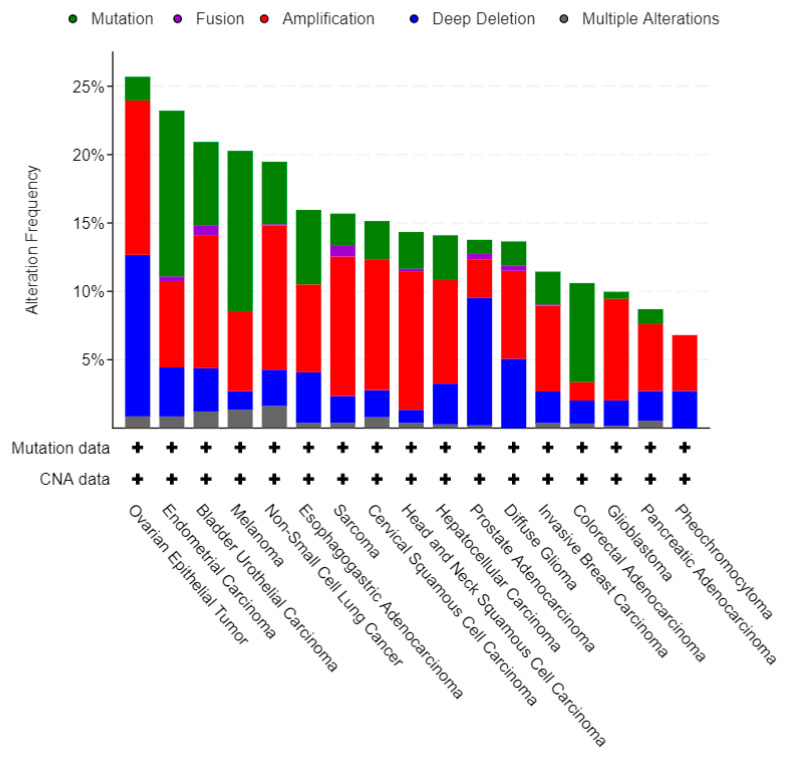
Global analysis of the cumulative alteration frequency of the queried genes per cancer type revealed that ovarian, endometrial, and bladder urothelial carcinomas are the three types of cancer with the higher frequency of mutation in the 5FU conversion system; only cases with >5% changes, occurring in a minimum of 100 total cases are represented; CNA, copy number alteration. Data retrieved from 32 studies including 10953 patients, TCGA PanCancer Atlas.

**Figure 4 biology-09-00265-f004:**
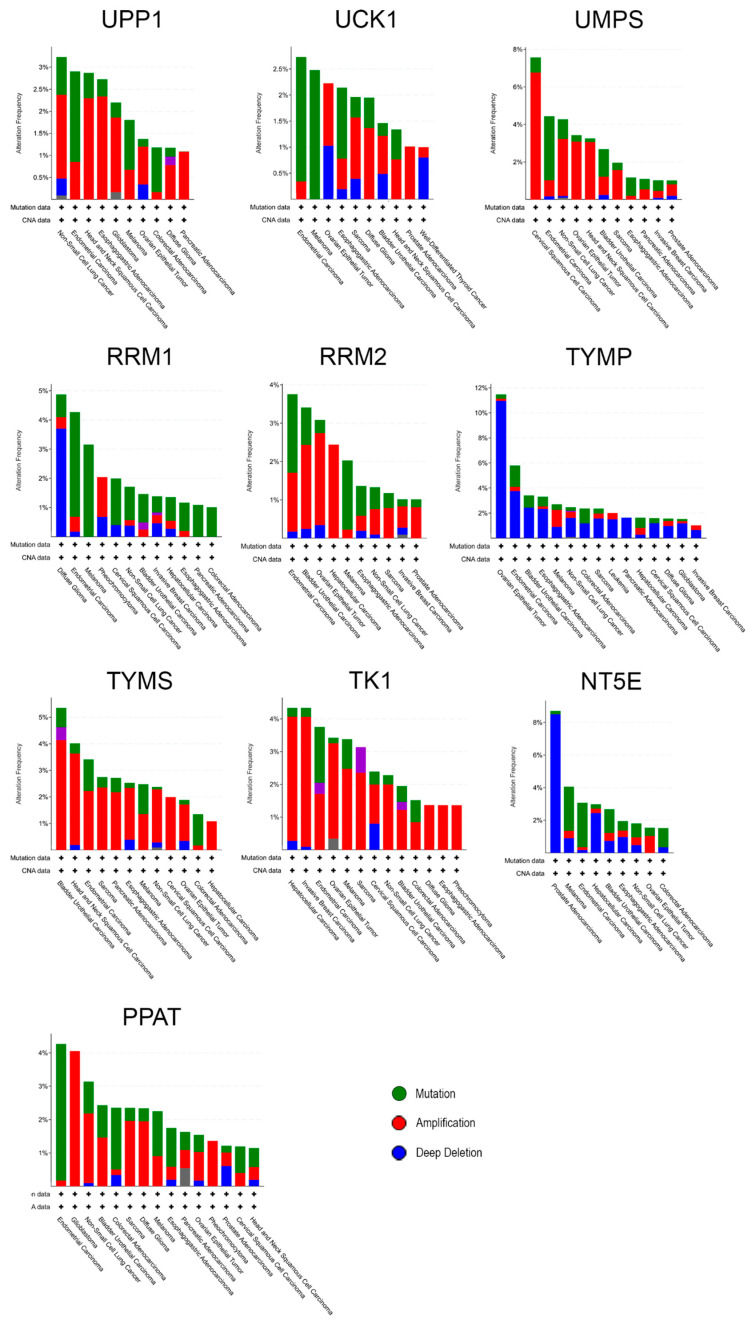
Individual analysis of the cumulative alteration frequency of the queried genes per cancer type; only cases with >1% changes, occurring in a minimum of 100 total cases, are represented; CNA, copy number alteration. Data retrieved from 32 studies including 10953 patients, TCGA PanCancer Atlas.

**Figure 5 biology-09-00265-f005:**
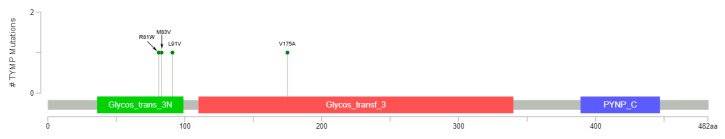
The aminoacidic sequence of *TYMP* gene; the protein domains are represented in green, the glycosyltransferase family helical bundle domain, in red the glycosyltransferase family a/b domain, and in blue the pyrimidine nucleoside phosphorylase C-terminal domain. Mutations are depicted as green lollipops with the annotation of the protein change above.

**Figure 6 biology-09-00265-f006:**
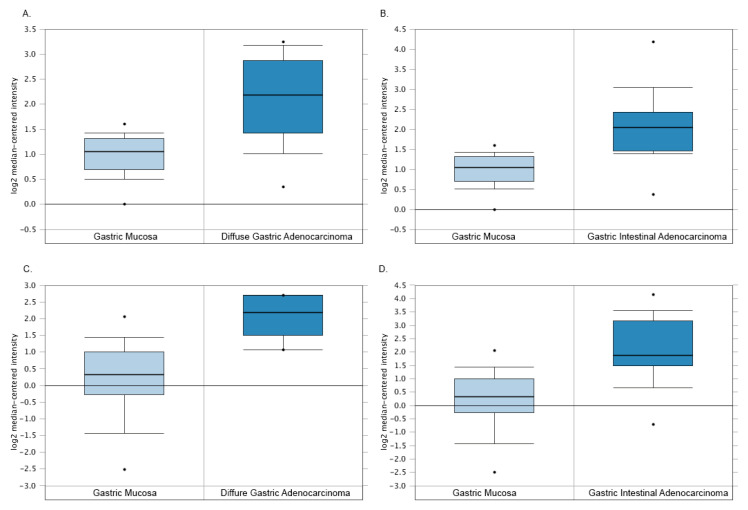
Comparison of TYMP gene expression in gastric mucosa with diffuse gastric adenocarcinoma (**A**,**C**) or gastric intestinal adenocarcinoma (**B**,**D**). *p* values < 0.0001 calculated by *t*-test analyzed through Oncomine.

**Figure 7 biology-09-00265-f007:**
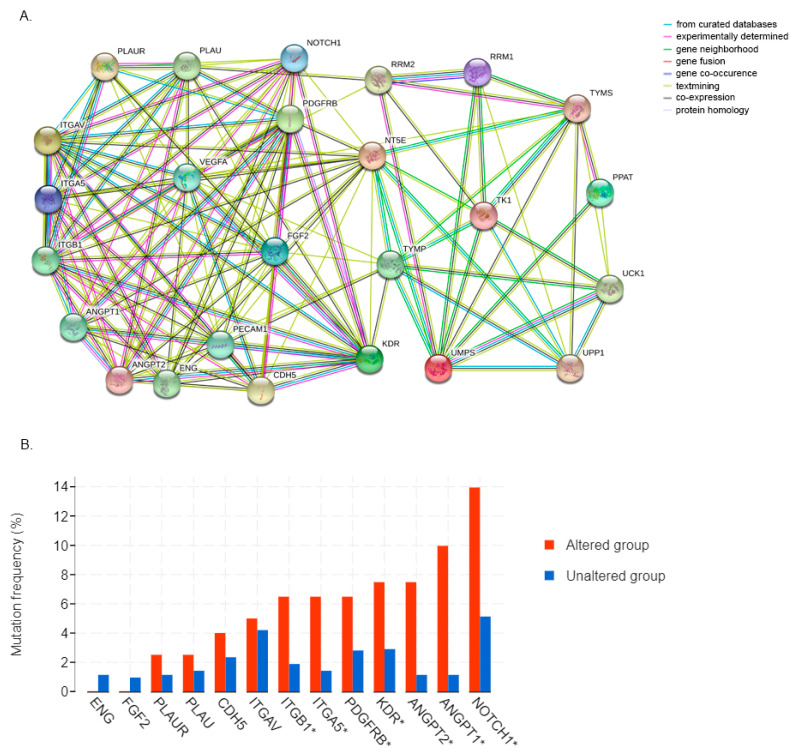
(**A**) A comprehensive map of the known and predicted interactions between 5FU converting system enzymes and several important proteins involved in tumor and physiological angiogenesis. (**B**) The analysis of the frequency of the mutations of the genes involved in angiogenesis in the altered and unaltered group described above. Briefly, the altered one includes all the patients with at least one of the 5FU metabolic genes mutated and the other one including all the “unaltered”. VEGFA was not reported in the cBioPortal database analyzing GC studies only; * *p* < 0.05.

**Table 1 biology-09-00265-t001:** Patients were divided into two groups: one with at least one mutation in the 5FU converting system defined as the altered group and one with the Wt allele version, the unaltered group. We observed significant modification in all the four examined parameters, i.e., the OS, the DF, the PFS, and the DSS in the group bearing mutations.

	Median Months Overall (OS)	Median Months Disease-Free (DF)	Median Months Progression-Free (PFS)	Median Months Disease-Specific (DSS)
Altered group	67.46	NA	46.98	95.57
Unaltered group	80.74	NA	65.00	152.02

**Table 2 biology-09-00265-t002:** We analyzed the main components of the 5FU conversion system, among 6 studies comprising 1365 patients regarding GC; SMF: somatic mutation frequency.

Gene	Protein	Mutation Frequency	# Mutations	Missense	Truncating	Inframe	SMF
*UPP1*	Uridine phosphorylase 1	2.6% (35)	3	2	1	0	0.4%
*UCK1*	Uridine-cytidine kinase 1	2.6% (35)	8	6	2	0	1.2%
*UMPS*	Uridine monophosphate synthetase *	1.4% (18)	6	2	4	0	1.0%
*RRM1*	Ribonucleotide reductase 1	1.4% (18)	5	5	0	0	1.1%
*RRM2*	Ribonucleotide reductase 2	1.4% (18)	5	4	1	0	0.8%
*TYMP*	Thymidine phosphorylase	2.8% (37)	4	4	0	0	0.7%
*TYMS*	Thymidylate synthase	1.4% (18)	1	1	0	0	0.2%
*TK1*	Thymidine kinase 1	1.4% (18)	1	1	0	0	0.1%
*NT5E*	Ecto-5′-nucleotidase	2.5% (33)	6	4	1	1	0.7%
*PPAT*	Phosphoribosyl pyrophosphate amidotransferase	2.2% (29)	7	5	2	0	1.3%
			N = 46	34 (73.9%)	11 (23.9%)	1 (2.2%)	

* OPRT catalytic domain is codified by UMPS; #: Number.

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
