# Peer review of "5-Fluorouracil Conversion Pathway Mutations in Gastric Cancer"

_biology, 2020, doi:10.3390/biology9090265_

Round 1
Reviewer 1 Report
Biagioni and colleagues employed available genomic data and software to detect alteration in 5-FU transformation pathways. The authors aimed to highlight the relevance of these metabolic routes in gastric cancer.
The manuscript is interesting; however, it is unclear which is the new "concept" that authors aims would like to discuss. I suggest to include this idea in the abstract, introduction and discussion sections of the manuscript.
In addition, I suggest authors to improve their analysis by comparing more than one statistical outcome at time (multivariate analysis). If authors they not have resources to perform such kind of analysis, I kindly ask to substitute/remove the word “meta-analysis” from all the manuscript, including the title. I consider this point important, as authors did not perform any analysis on results obtained from cBioportal.
Author Response
Biagioni and colleagues employed available genomic data and software to detect alteration in 5-FU transformation pathways. The authors aimed to highlight the relevance of these metabolic routes in gastric cancer.
The manuscript is interesting; however, it is unclear which is the new "concept" that authors aims would like to discuss. I suggest to include this idea in the abstract, introduction and discussion sections of the manuscript.
We thank the reviewer for the appreciation and suggestions to improve our manuscript quality. We expressed in the Discussion and Conclusion sections the aim of our analytic work as “We do believe that the evaluation of TYMP, as the key enzyme of the 5FU pathway, and being the one which is often find mutated in GC, might improve the choice for the best therapy regimen. Indeed, the use of a wrong treatment not only causes cytotoxicity without any beneficial effects, but might also select a pool of chemoresistant cells that may gain a selective advantage to access blood and lymphatic vessels to metastasize by stimulating tumor angiogenesis.”. The main focus of our manuscript is pointing out the attention of the scientific community to the research of reliable markers which might reflect the chemosensitivity status of the patients and may be evaluable during the therapy. We agree that the concept can be better stated, therefore we add a sentence to the Abstract, Introduction and Discussion sections accordingly.
In addition, I suggest authors to improve their analysis by comparing more than one statistical outcome at time (multivariate analysis). If authors they not have resources to perform such kind of analysis, I kindly ask to substitute/remove the word “meta-analysis” from all the manuscript, including the title. I consider this point important, as authors did not perform any analysis on results obtained from cBioportal.
Thank you for your precious suggestion. Unfortunately, it is currently not possible to perform a significant multivariate analysis exploiting only cBioportal as such a tool classifies patients with a mutation in the queried gene as “altered”, which might results either in an increased or decreased expression, while all the others as “non-altered”. Currently, in line with the fixed deadline we are unable to perform further analysis on RNA seq or IHC specimens, therefore following your request we excluded the word “Meta-analysis” from the title.
Reviewer 2 Report
This manuscript of Biagioni and colleagues reports on the Fluorouracil conversion pathway mutations in gastric cancer: a meta-analysis. The study might be of relevance for gastric cancer. However, the manner in which the results are reported is unclear and focuses on clinical phenomena, and it is unclear what role the in-depth molecular mechanisms might play. This article has certain scientific value and significance, but there are still many needs to be strengthened and improved for this manuscript. However, some issues need improvement and/or clarification before publication in “Biology”.
1. There is a lack of mechanism-related research, and the current preliminary evidence only focuses on clinical data. Although there are a larger number of cases, the author should thoroughly study its possible mechanism generation. This will have the opportunity to provide more scientific evidences.
2. The manuscript is technically sound and well written. The concept of the work is valid; however, the paper requires a major effort to present the data adequately. However, thus these results could be rather a preliminary one.
3. In the Materials and Methods and Results, the description of Materials and Methods is too concise, the author should greatly increase the details.
4. In the Results (Fig 2 and 4), the Figs should be reconsidered and greatly improve the quality, it becomes clear to the reader what each of the different columns represent. The quality is poor now, not easy to read.
5. Although, the authors compare TYMP gene expression in gastric mucosa with diffuse gastric adenocarcinoma or gastric intestinal adenocarcinoma using Oncomine database. However, authors should provide additional databases or use human patient tissues for further double verification.
6. The author should also provide a comparison of TYMP gene expression in the gastric mucosa of diffuse gastric adenocarcinoma or gastrointestinal adenocarcinoma with patient survival.
7. In the Results, authors should explore the TYMP gene correlation between gastric cancer markers, proliferating cell nuclear antigen (PCNA) and apoptosis caspase-3 expression and available inflammatory indexes in gastric cancer patients.
8. In the Discussion, the discussion mostly covers previously published results. I would recommend the authors to concentrate more on how their own results complement/contradict the existing data. it is necessary to introduce a possible mechanism and the cause of the generation of gastric cancer or other tumors, which can provide more information to the reader's reference or step-by-step study to clarify the mechanism of this diseases.
9. The manuscript contains some odd use of English and should be proofed by a native speaker.
10. This article is more suitable for submission to cancer epidemiology related journals.
Author Response
This manuscript of Biagioni and colleagues reports on the Fluorouracil conversion pathway mutations in gastric cancer: a meta-analysis. The study might be of relevance for gastric cancer. However, the manner in which the results are reported is unclear and focuses on clinical phenomena, and it is unclear what role the in-depth molecular mechanisms might play. This article has certain scientific value and significance, but there are still many needs to be strengthened and improved for this manuscript. However, some issues need improvement and/or clarification before publication in “Biology”.
- There is a lack of mechanism-related research, and the current preliminary evidence only focuses on clinical data. Although there are a larger number of cases, the author should thoroughly study its possible mechanism generation. This will have the opportunity to provide more scientific evidences.
We appreciate the reviewer’s suggestion to focus the attention on the mechanisms influencing the mutation pattern of gastric cancer and the ones required for chemoresistance occurrence. However, this is not the topic of our manuscript as we aimed only to focus on the mutation pattern of the 5-Fluorouracil metabolic pathway and its implications in patients’ OS. The generation of mutations, as well as the occurrence of chemoresistance, would require to be analyzed in different manuscripts. As a multidisciplinary unit, we are to date investigating the TYMP-related mechanism occurring during 5-FU chemoresistance in Caucasian gastric cancer but as stated by the MDPI instruction for the author the Concept Papers should not contain new experimental data.
- The manuscript is technically sound and well written. The concept of the work is valid; however, the paper requires a major effort to present the data adequately. However, thus these results could be rather a preliminary one.
Thanks for your appreciation. As stated above, our article has been submitted as a “Concept Paper” including information taken by one of the major databases (cBioportal, TGCA) and discussing how the 5-Fluorouracil related enzymes might influence patients’ OS when mutated. We do hope to be able in the future to deepen such topics with experimental results in a full scientific article but as reported in the MDPI instruction for the author the Concept Papers should not contain new experimental data.
- In the Materials and Methods and Results, the description of Materials and Methods is too concise, the author should greatly increase the details.
We agree with the reviewer the Material and Methods is very concise due to the lack of experimental data produced, but we do believe that we reported all the information needed to replicate the analysis. Moreover, in each figure captions were reported the studies selected by cBioportal and Oncomine and the limitations that we used to obtain high quality and significant data. However, suggestions to improve this section are welcome.
- In the Results (Fig 2 and 4), the Figs should be reconsidered and greatly improve the quality, it becomes clear to the reader what each of the different columns represent. The quality is poor now, not easy to read.
Data were downloaded directly from cBioportal and composed through the suite Photoshop 20.0.0 20180920.r.24 2018/09/20: 1193433 x64 with 300 dpi render. However, following your precious suggestion we replaced manually the labels in both figures, which might result a bit blurry in the final .doc document and in particular Figure 4 was organized in a vertical landscape to increase the column and the label size.
- Although, the authors compare TYMP gene expression in gastric mucosa with diffuse gastric adenocarcinoma or gastric intestinal adenocarcinoma using Oncomine database. However, authors should provide additional databases or use human patient tissues for further double verification.
Currently, as stated in the Discussion section, there are few data available on gastric cancer and often the lack of clinical information made such results unreliable. For example, being TYMP a crucial gene in the response to 5-Fluorouracil, the use of adjuvant therapies might influence its expression but such data is not always reported in several databanks. Due to COVID-19 limitations, we are currently unable to provide human patient tissues for IHC analysis and the restriction regarding the Concept Papers imposes not to use new empirical data. However, the data obtained by Oncomine refers to 2 separate studies provided by Cho JY et al. [Cho, J.Y.; Lim, J.Y.; Cheong, J.H.; Park, Y.-Y.; Yoon, S.-L.; Kim, S.M.; Kim, S.-B.; Kim, H.; Hong, S.W.; Park, Y.N.; et al. Gene expression signature-based prognostic risk score in gastric cancer. Clin. Cancer Res. 2011, 17, 1850–1857, doi:10.1158/1078-0432.CCR-10-2180.] and D’Errico M et al. [D’Errico, M.; de Rinaldis, E.; Blasi, M.F.; Viti, V.; Falchetti, M.; Calcagnile, A.; Sera, F.; Saieva, C.; Ottini, L.; Palli, D.; et al. Genome-wide expression profile of sporadic gastric cancers with microsatellite instability. Eur. J. Cancer 2009, 45, 461–469, doi:10.1016/j.ejca.2008.10.032.]. Moreover, as reported in reference 50, Kimura H et al. [Kimura, H.; Konishi, K.; Kaji, M.; Maeda, K.; Yabushita, K.; Miwa, A. Correlation between expression levels of thymidine phosphorylase (dThdPase) and clinical features in human gastric carcinoma. Hepatogastroenterology 2002, 49, 882–886.] were able to observe the same phenomenon.
- The author should also provide a comparison of TYMP gene expression in the gastric mucosa of diffuse gastric adenocarcinoma or gastrointestinal adenocarcinoma with patient survival.
We thank the reviewer for the precious suggestion as such data might help to improve the knowledge of TYMP status in the two different histotypes of gastric cancer. Unfortunately, there is no data available to date to accomplish such a request. As reported in the manuscript, even though we analyzed 6 different studies comprising 1365 gastric cancer patients, mutations in TYMP gene were reported in 37 samples which are unreliable for fulfilling the reviewer’s request with significant data. Moreover, cBioportal is a tool that classifies patients with a mutation in the queried gene as “altered”, which might result either in an increased or decreased expression, while all the others are classified as “non-altered”, which is the real motivation why we decided to exploit Oncomine too.
- In the Results, authors should explore the TYMP gene correlation between gastric cancer markers, proliferating cell nuclear antigen (PCNA) and apoptosis caspase-3 expression and available inflammatory indexes in gastric cancer patients.
Excluding CEA, CA 19-9, and CA 72-4, which are tumor markers commonly used in several neoplasms, we focused the required analysis on HER2, EGR1, KLF5, CA IX, Ki67, E-Cadherin, CK7/20, PCNA, CASP3 and IL8, and NOS2 as inflammatory markers. Such analysis, included in Figure 7 and discussed in the relative Result section, demonstrated a correlation occurring significantly only for KLF5, CAIX and Ki67 between the two selected cohorts. Exploring such correlations taking into consideration only the TYMP mutated cohort did not produce any result due to the limited number of TYMP mutated samples.
- In the Discussion, the discussion mostly covers previously published results. I would recommend the authors to concentrate more on how their own results complement/contradict the existing data. it is necessary to introduce a possible mechanism and the cause of the generation of gastric cancer or other tumors, which can provide more information to the reader's reference or step-by-step study to clarify the mechanism of this diseases.
The cause of the generation of gastric cancer as well as the generation of tumor-related mutations were purposely not included in the submitted manuscript as they deserve dedicated reviews/articles. We partially reported some of the requested topics in a recent review article [Biagioni A, Skalamera I, Peri S, et al. Update on gastric cancer treatments and gene therapies. Cancer Metastasis Rev. 2019;38(3):537-548. doi:10.1007/s10555-019-09803-7] which was already included in the References. Moreover, we do believe that our manuscript, discussing the mutations occurring in the 5FU conversion system, should not include general concepts such as the gastric cancer tumorigenesis and the generation of tumor-related mutations, which may result misleading for the readers that should be focused in the postulated importance of the 5-FU conversion pathway.
- The manuscript contains some odd use of English and should be proofed by a native speaker.
Thank you for your suggestion. The manuscript was entirely revised to correct any use of odd words and grammatical mistakes. If the reviewer is not satisfied with the revisions made, we will ask MDPI English service to improve the manuscript, as we are currently unable to exploit a native speaker proofread.
- This article is more suitable for submission to cancer epidemiology related journals.
We agree with the reviewer that there are many journals suited for our article however, the author intended to combine the crucial clinical data obtained by large databases and to discuss how the mutations in the 5FU conversion pathway might impact not only the OS but also the angiogenic process.
Round 2
Reviewer 2 Report
The author has made good corrections and responses. I suggest that this version could be published.
Author Response
We appreciated the reviewer's response.
Many thanks for your efforts in improving our manuscript.
This manuscript is a resubmission of an earlier submission. The following is a list of the peer review reports and author responses from that submission.
Round 1
Reviewer 1 Report
In this manuscript entitled “5-Fluorouracil conversion pathway mutations in gastric cancer”, Alessio Biagioni and colleagues describe that mutations in ten critical enzymes involved in 5-Fluorouracil metabolism, were led to a poor prognosis with reduced overall survival. The results seem valuable in finding a group of gene mutations that lead to a poor prognosis. However, the following concerns need to be addressed.
- One major concern is regarding the study design and data analysis. The authors claim “5FU system mutations lead to a poor prognosis”. However, the results were obtained by combining all mutations in 10 genes. The mutation rate is very low in each individual genes; the range from 1.1% to 2.4%, an average 1.7%. The author's approach means the patients were grouped into altered when anyone of 10 genes mutation. The total of the altered group increases to 14%, and it means most of the patients only mutation one or two genes. The more robust approach and I would suggest to do is to analyze survival in individual genes and further study the combined mutation genes in the same patients.
- In the results 3.3, “Individual analysis of mutation frequency”. The authors claim that TYMP and TYMS had the higher percentage of mutation. However, the TYMS (1.7%) is the fifth mutation gene in my analysis. The top 4 mutation rate genes are TYMP (2.4%), TK1 (1.9), UMPS (1.8%), and PPAT (1.7%).
- In the results 3.4, “5FU conversion system mutations analysis in GC”. The authors claim that being well-known the discordances between Asian and Caucasian populations’ prognosis in GC. So the authors want to understand better the role of the 5FU conversion system mutations in GC. However, the study indicates that white patients had poorer cancer-specific survival for prostate cancer, breast cancer, stomach cancer, pancreatic cancer, lung cancer, liver and/or IHBD cancer, and colorectal cancer. White patients also had lower overall survival than Asian patients for prostate cancer, breast cancer, stomach cancer, lung cancer, liver and/or IHBD cancer, and colorectal cancer (Zhang C, et al. JAMA Network Open. 2020. PMID: 32267515). Gastric cancer neither the only one cancer that prognosis different in different races nor the top 1 mutation rate cancer. The reason for selecting gastric cancer for further study may not be reasonable.
- In the results 3.5, “The paradoxical TYMP expression in GC”. The authors show the expression of TYMP in tumor type of gastric cancer higher than normal type. However, the TYMP expression higher in cancer type may well study. Copy from Tabata et al., Cell Reports, 2017. TYMP (TP) expression in various malignant tumors is higher than in adjacent non-neoplastic tissues (Takebayashi et al., 1996b), and poor prognosis is associated with TP-positive versus TP-negative colon and differentiated gastric carcinomas (Shimaoka et al., 2000; Takebayashi et al., 1996a).
Author Response
Thanks for your valuable suggestions, we put every effort to improve our manuscript following your concerns and requests.
One major concern is regarding the study design and data analysis. The authors claim “5FU system mutations lead to a poor prognosis”. However, the results were obtained by combining all mutations in 10 genes. The mutation rate is very low in each individual genes; the range from 1.1% to 2.4%, an average 1.7%. The author's approach means the patients were grouped into altered when anyone of 10 genes mutation. The total of the altered group increases to 14%, and it means most of the patients only mutation one or two genes. The more robust approach and I would suggest to do is to analyze survival in individual genes and further study the combined mutation genes in the same patients.
We agree with the reviewer that such topic would need more accurate analyses and more precise data but even large databases, such cBioPortal and Oncomine, lack sometimes relevant information about each patient, like the chemotherapy regimen, the BMI and the response to chemo- or radiotherapy. Moreover, analyzing the survival rates of patients with amplification vs deletion of a single gene would require larger cohorts and bigger studies. Here, we limited to exploit a combined approach in order to have more robust data leading to significant results: we included all the patients with at least one mutation in the 5FU pathway in a single group (the altered) vs a group composed by patients without 5-FU-pathway related mutations (the unaltered), to analyze the survival rate and the mutation frequency among several kind of cancers, while gene mutation rate was evaluated singularly. As the reviewer suggested the mutation rate is very low but it is relative to 6 studies including 1365 patients, meaning that at least one of such mutations is findable in 211 (15%) patients of the total examined. Moreover, while in the first part of our study we extended the analysis to all kind of cancers included in the cBioPortal database, as reported in paragraph 3.1 and in Figure 2 legends, the analysis of 5FU metabolism-related mutation rate was focused only on gastric cancer. However, even if we would repeat the mutation rate analysis on all kind of cancers, expanding the number of patients to 10953, the rates will not change in a significant way (UPP1 1.4%, UCK1 1.1%, UMPS 1.8%, TYMP 2.4%, TYMS 1.7%, RRM1 1.4%, RRM2 1.2%, PPAT 1.8%, TK1 1.9% and NT5E 1.7 – comprising a total of 1488 patients with at least one altered gene, which is about the 14%). To improve the quality of our manuscript in accordance with the reviewer observation we included in the supplementary material, a co-occurrence mutation analysis and the overall survival analysis of individual genes, which were consequently discussed in the text.
In the results 3.3, “Individual analysis of mutation frequency”. The authors claim that TYMP and TYMS had the higher percentage of mutation. However, the TYMS (1.7%) is the fifth mutation gene in my analysis. The top 4 mutation rate genes are TYMP (2.4%), TK1 (1.9), UMPS (1.8%), and PPAT (1.7%).
We thank the reviewer for its kind observation. We corrected the mistake in the sentence.
In the results 3.4, “5FU conversion system mutations analysis in GC”. The authors claim that being well-known the discordances between Asian and Caucasian populations’ prognosis in GC. So the authors want to understand better the role of the 5FU conversion system mutations in GC. However, the study indicates that white patients had poorer cancer-specific survival for prostate cancer, breast cancer, stomach cancer, pancreatic cancer, lung cancer, liver and/or IHBD cancer, and colorectal cancer. White patients also had lower overall survival than Asian patients for prostate cancer, breast cancer, stomach cancer, lung cancer, liver and/or IHBD cancer, and colorectal cancer (Zhang C, et al. JAMA Network Open. 2020. PMID: 32267515). Gastric cancer neither the only one cancer that prognosis different in different races nor the top 1 mutation rate cancer. The reason for selecting gastric cancer for further study may not be reasonable.
In the current study we did not take under consideration parameters such as race, sex or ethnicity. All the data and the patients were selected only by their cancer type and 5-FU pathway mutation pattern. We stated in paragraph 3.4 that the genetic compartment plays a fundamental role in the progression of gastric cancer and as reported by Jia, F. et al. [Discordance of Somatic Mutations Between Asian and Caucasian Patient Populations with Gastric Cancer. Mol Diagn Ther 2017, 21, 179–185] different prognoses, based on race/ethnicities, rely on genetic and molecular discordances and not only on environmental causes (such as alcohol and food habits, in the particular case of gastric cancer). We know that race is not a selective marker of prognosis only in gastric cancer but we decided to proceed in our analysis as we are currently studying gastric adenocarcinomas, due to the extensive background of Prof. Cianchi and Dr. Staderini, who are surgeons for the digestive system at the AOUC Careggi Hospital and we do believe that TYMP, especially for gastric cancer, might play a major role in chemoresistance and therefore will deserve to be further investigated experimentally. We therefore added the kindly suggested reference to the manuscript.
In the results 3.5, “The paradoxical TYMP expression in GC”. The authors show the expression of TYMP in tumor type of gastric cancer higher than normal type. However, the TYMP expression higher in cancer type may well study. Copy from Tabata et al., Cell Reports, 2017. TYMP (TP) expression in various malignant tumors is higher than in adjacent non-neoplastic tissues (Takebayashi et al., 1996b), and poor prognosis is associated with TP-positive versus TP-negative colon and differentiated gastric carcinomas (Shimaoka et al., 2000; Takebayashi et al., 1996a).
We agree with the reviewer that TYMP expression would deserve to be studied in deep but with the present manuscript we only aimed to focus on the mutation profile of the queried gene, stating that although TYMP is commonly deleted, we evidenced a higher expression in both two different types of gastric cancer. However, more importantly, to date there are not studies available on TYMP level in 5FU chemoresistant gastric cancer patients, a topic, we hope, will be studied in the very next future. We added the reference kindly suggested by the reviewer with a brief explanation in the manuscript.
Finally, the manuscript was entirely revised both in the content and the grammar.
Reviewer 2 Report
5-Flurouracil is a common anti-cancer drug used in the treatment of many cancers. In this study, using cBioportal analysis, the authors investigated the components of the three major 5FU transformation pathways and suggested that poor prognosis and reduced overall survival observed is caused by deletion in TYMP gene and amplification of TYMS. The authors suggested that evaluation of TYMP may improve the choice for the best therapy regimen.
Comments:
- This is an interesting concept paper but requires extensive editing for grammar and choice of words throughout the text. For example, the authors use the word “leaded” in 4 different places- line 17, line 22, line 104, and line 262. In my opinion, it is not a correct word. May be the authors meant “lead”?
- My suggestion to the reviewers is to revise and re submit the paper.
Author Response
This is an interesting concept paper but requires extensive editing for grammar and choice of words throughout the text. For example, the authors use the word “leaded” in 4 different places- line 17, line 22, line 104, and line 262. In my opinion, it is not a correct word. May be the authors meant “lead”?
My suggestion to the reviewers is to revise and re submit the paper.
Thank you so much for your valuable comments. We deeply revised the paper, edited the grammar, changed the most controversial words and corrected the errors found. Finally, a supplementary material section was added to clarify some uncertainty points.
Round 2
Reviewer 1 Report
In your manuscript you describe RRM1 might have a positive impact on the OS, mutations in UPP1 and PPAT are strongly
156 associated with a worse prognosis (Supplementary Figure 1).
My questions are
(1) Why the unaltered group always the same after you change the alter gens?
(2) You also have to show the p-value to present that is a significant change.
The reason for selecting gastric cancer for further study still may not be reasonable. I know that you may be more concerned or interested in gastric cancer. However, you should either have a more consistent story or only analyze gastric cancer initially. The gap from pan-cancer to gastric cancer still exists.
Reviewer 2 Report
5-Flurouracil is a common anti-cancer drug used in the treatment of many cancers. In this study, using cBioportal analysis, the authors investigated the components of the three major 5FU transformation pathways and suggested that poor prognosis and reduced overall survival observed is caused by deletion in TYMP gene and amplification of TYMS. The authors suggested that evaluation of TYMP may improve the choice for the best therapy regimen.
Comments:
- The current version is not revised enough to make it a scientifically sound paper. It is slightly better than the previous version, changes that have been made to the original version are only cosmetic. I am still unable to read and understand the scientific meanings conveyed in the sentences.
- Line 43 and 44 reads : While FdUMP inhibits the DNA synthesis acting on TYMS, FUMP is the main responsible for gene expression blockade”. This is a poorly written sentence. A better and meaningful sentence can be “ While FdUMP inhibits DNA synthesis through its inhibitory action on TYMS, FUMP is largely responsible for the inhibition of gene expression though its incorporation into mRNAs.
- Line 44 to Line47 reads “Another conversion way is mediated by the action of the thymidine phosphorylase (TYMP) which is able to convert 5FU into fluorodeoxyuridine (FdU) and then transformed to FdUMP thanks to the thymidine kinase1 (TK1) in a process called TP-TK pathway”. I want to remind the reviewers that authors do not use the words such as ‘thanks” in the scientific papers in the main body of the text.
- Line 76 to 79- reads “Actually the gold standard, following international guidelines, recommends a first line approach with a therapy containing a platinum agent (cisplatin or oxaliplatin) and fluoropyrimidines (5-fluorouracil or capecitabine) in patients HER2 negative, otherwise trastuzumab is added for HER2 positive patients” From this sentence it is not clear what the authors are trying to convey. A better and more meaningful sentence could be “According to the international guidelines, the first line approach for therapy would involve treating HER2 negative patients with platinum-based drugs; in contrast, in HER2 positive patients are to be treated with transtuzumab”. Please convey the scientific meaning very carefully through out the text.
- Line 137- states that “As 5FU is a one of the most diffuse components in many kinds of cancers……” . Most diffuse components? Please clarify what that means.
- Line 237- Legends for figure 6- states that “TYMP gene expression comparing gastric mucosa Vs diffuse gastric adenocarcinoma or gastric intestinal type carcinoma” A better sentence could be- “Comparison of the TYMP gene expression in gastric mucosa with diffuse gastric adenocarcinoma (A and C) or gastric intestinal adenocarcinoma (B and D). Is this what the authors mean?
- Discussion is poorly written. Sentences are not clear.